# Clinical CDK4/6 inhibitors induce selective and immediate dissociation of p21 from cyclin D-CDK4 to inhibit CDK2

Lindsey R. Pack[1,2], Leighton H. Daigh [1,2], Mingyu Chung[1] & Tobias Meyer [1✉]

Since their discovery as drivers of proliferation, cyclin-dependent kinases (CDKs) have been considered therapeutic targets. Small molecule inhibitors of CDK4/6 are used and tested in clinical trials to treat multiple cancer types. Despite their clinical importance, little is known about how CDK4/6 inhibitors affect the stability of CDK4/6 complexes, which bind cyclins and inhibitory proteins such as p21. We develop an assay to monitor CDK complex stability inside the nucleus. Unexpectedly, treatment with CDK4/6 inhibitors—palbociclib, ribociclib, or abemaciclib—immediately dissociates p21 selectively from CDK4 but not CDK6 complexes. This effect mediates indirect inhibition of CDK2 activity by p21 but not p27 redistribution. Our work shows that CDK4/6 inhibitors have two roles: non-catalytic inhibition of CDK2 via p21 displacement from CDK4 complexes, and catalytic inhibition of CDK4/6 independent of p21. By broadening the non-catalytic displacement to p27 and CDK6 containing complexes, next-generation CDK4/6 inhibitors may have improved efficacy and overcome resistance mechanisms.

[1] Department of Chemical and Systems Biology, Stanford University, Stanford, CA, USA. [2] These authors contributed equally: Lindsey R. Pack, Leighton H. Daigh. ✉email: tobias1@stanford.edu

Cell-cycle entry in early G1 phase relies on the activation of CDK4 and CDK6 by binding of cyclin D, while progression through G1 and entry into S phase further relies on CDK2 activation by binding of cyclin E. The activity of both types of dimeric cyclin-CDK complexes can be inhibited by the CDK Interacting Protein/Kinase Inhibitory Protein (CIP/KIP) family of protein inhibitors. These inhibitors—p21, p27, and p57—bind to cyclin-CDK dimers to form inactive trimeric complexes[1].

Clinical inhibitors of CDK4/6 kinases have been approved as a therapeutic for estrogen receptor-positive breast cancers, and more than 100 clinical trials are currently evaluating CDK4/6 inhibitors in cancer treatment[2–5]. Resistance to CDK4/6 inhibitors occurs via multiple mechanisms, including mutation of *RB1*, increased activity of mitogenic signaling, upregulation of CDK2 activity, or suppression of the levels of CIP/KIP proteins, p21 and p27[5–10]. Interestingly, amplification and activation of CDK4 is associated with enhanced sensitivity to CDK4/6 inhibitors[11,12], while amplification of CDK6 can overcome the suppressive role of CDK4/6 inhibitors[6,13–16]. Despite their clinical importance, few studies have explored how the binding of clinical CDK4/6 inhibitors regulates the dimeric and trimeric CDK4/6 complexes or if differential responses to CDK4/6 inhibitors occur between CDK4 complexes vs. CDK6 complexes.

Recently, Guiley et al. reported that clinical CDK4/6 inhibitors predominantly bind to monomeric CDK4 and CDK6 in cells, proposing that these small-molecule inhibitors interfere with folding but do not bind to and inhibit CDK4/6 dimers and trimers in cells[17]. Instead, they suggest that clinical CDK4/6 inhibitors act by a mechanism that requires days of treatment. This interpretation cannot readily be reconciled with another recent study demonstrating that treatment of late G1 cells with clinical CDK4/6 inhibitors causes Rb dephosphorylation within 15 min[18]. The observed rapid dephosphorylation of Rb and inhibition of cell-cycle progression implies instead that clinical CDK4/6 inhibitors can act by direct catalytic inhibition of CDK4/6 kinase activity.

Here, we develop a method to measure the stability of protein complexes in live cells to understand if and how the function of dimeric and trimeric CDK complexes is regulated within the nucleus by clinical CDK4/6 inhibitors. We show CDK4/6 inhibitors rapidly access already formed cyclin D-CDK4-p21 trimeric complexes to lower p21 affinity without preventing the formation of dimeric and trimeric complexes. Unexpectedly, treatment with clinical CDK4/6 inhibitors results in an immediate accelerated dissociation of p21 only from cyclin D-CDK4-p21 complexes but not from cyclin D-CDK6-p21 complexes or complexes containing p27. Moreover, we find that the dissociation of p21 from cyclin D-CDK4-p21 complexes induced by clinical CDK4/6 inhibitors results in rapid inhibition of CDK2 kinase activity by binding of the released p21. Together, we show that clinical CDK4/6 inhibitors act immediately in cells to inhibit cell-cycle entry in two ways: a CDK4-restricted non-catalytic role to indirectly inactivate CDK2 complexes via p21 redistribution and a catalytic role to directly inactivate CDK4 and CDK6 kinase activity.

## Results

**Dissociation of protein complexes measured live in the nucleus of cells.** Binding parameters of CDK4/6 complexes are challenging to measure in vitro due to the difficulty to express and purify CDK4/6 kinases that have a necessary T-loop phosphorylation, which is required for activation by separately purified cyclin D protein[19–22]. As a consequence, how the stability and activity of dimeric and trimeric CDK4/6 complexes are regulated by clinical CDK4/6 inhibitors is not well understood. Here, we developed a method to analyze CDK4/6 complexes in their physiological

nuclear setting by measuring the dissociation of cyclins and CDK inhibitors from CDK4/6 in the nucleus of live cells. The method measures binding under equilibrium conditions, which allows for accurate quantification of binding properties[23]. Our approach combines stable targeting of CDKs to the inner nuclear membrane with fluorescence recovery after photobleaching (FRAP) analysis to measure the time course of dissociation of cyclins or CDK inhibitors and thus the stability of dimeric and trimeric CDK complexes.

To tether CDKs to the nuclear periphery, we fused CDKs to a lamin A protein lacking 50 amino acids near the C terminus (Δ50 lamin A), which was shown to stably insert into the inner nuclear membrane[24]. To test if Δ50 lamin A can be used as a stable tether for CDKs, we designed a construct fusing CDK6-GFP to the Δ50 lamin A coding sequence (Fig. 1a). Transient expression in human retinal pigment epithelial cells (RPE-1) showed the expected localization of the fusion protein to the nuclear periphery (Fig. 1b). FRAP analysis demonstrated that the tethered CDK6 had minimal turnover, with little fluorescence recovery occurring over 15 min of imaging (Fig. 1b, c). Similar results were observed for CDK4-GFP-Δ50 lamin A. Limited fluorescence recovery of tethered CDKs demonstrates that turnover and diffusion of CDKs will not confound FRAP-based measurements of dissociation rates of CDK binding proteins. Thus, the method allows for measurements of the stability of CDK complexes in the nucleus by monitoring the dissociation time of cyclins and CDK inhibitors from tethered CDKs.

Two families of CDK inhibitors repress CDK activity—the CIP/KIP family, which binds and inhibits CDK4/6 and CDK2 when in complex with cyclins[1,25,26], and the INK4 family, which binds monomeric CDK4/6 and allosterically inhibits the binding of cyclin D[27,28]. To determine if CDK binding proteins could still access the tethered CDKs, we co-transfected a plasmid encoding mRuby3-tagged p16, an INK4 family member, which binds CDK6 with high affinity to form a dimeric complex[20]. Indeed, expressed p16 was highly enriched at the nuclear periphery, showing that p16 can access and bind the tethered CDK6 (Fig. 1d). We confirmed that enrichment required CDK6 interactions, as no enrichment of p16 or other fluorescently tagged CDK binding partners was observed in cells expressing the GFP-Δ50 lamin A lacking CDK6 (Supplementary Fig. 1a).

We determined the strength of the CDK6-p16 interaction by measuring the mRuby3-p16 dissociation from tethered CDK6 using FRAP (Fig. 1e). Previous studies using surface plasmon resonance reported a very tight binding affinity of p16 to CDK6 in the sub-nanomolar range[20]. To prevent cell movement during the expected slow recovery of p16 fluorescence, we pre-treated cells with nocodazole, to prevent microtubule-mediated movement, and a drug cocktail that stabilizes the actin cytoskeleton[29]. FRAP analysis of CDK6-bound p16 showed minimal recovery of the photobleached p16 signal after 15 min, consistent with a high affinity interaction between the two proteins in the nuclear environment (Fig. 1f). Similar results were observed for p16 bound to tethered CDK4. To examine if the tether system can be used to measure lower affinity interactions, we measured the recovery of a cancer-associated p16 mutant, p16 (D84N), which has a greatly reduced binding affinity towards CDK6[20,30]. p16 (D84N) still bound the tethered CDK6 construct at the nuclear periphery, demonstrating that some binding activity was retained. However, recovery of p16 (D84N) fluorescence following photobleaching occurred rapidly, indicating a greatly increased dissociation-rate consistent with the reduced binding affinity of p16 (D84N) for CDK6 (Fig. 1g and Movie S1). Moreover, p16 (D84N) recovery was unaffected by the cytoskeleton-stabilizing drugs, indicating that our measurements were not affected by actin stabilization or microtubule inhibition (Supplementary Fig. 1b).

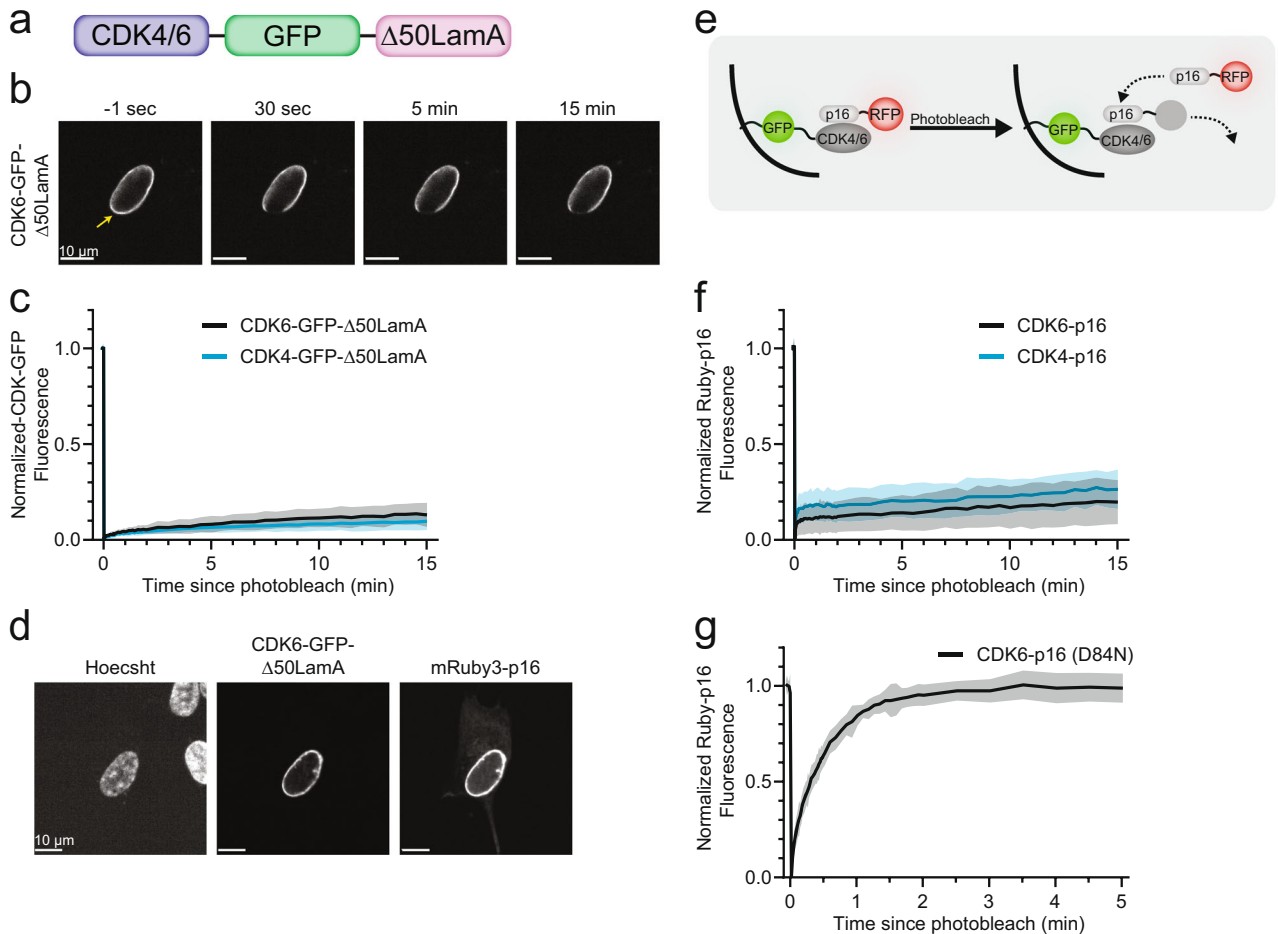

**Fig. 1 Stable tethering of CDKs to inner nuclear membrane enables FRAP-based measurement of CDK complex stability. a** Schematic of engineered CDK constructs. CDK4 or CDK6 were conjugated at the C-terminus with GFP and Δ50 lamin A, which stably tethers the CDKs to the inner nuclear membrane. **b** Timelapse confocal images of CDK6-GFP-Δ50 lamin A fusion construct transiently transfected into RPE-1 cells. Cells were followed for 15 min after photobleaching to measure recovery of the GFP fluorescence signal. **c** Control experiment, showing a very slow recovery of CDK4- ($n = 18$ cells, 2 biological replicates) and CDK6-GFP-Δ50 lamin A ($n = 11$ cells, 2 biological replicates) fluorescence following photobleaching. **d** Confocal fluorescence images of RPE-1 cells co-transfected with CDK6-GFP-Δ50 lamin A and mRuby3-p16 and stained with Hoechst. **e** Schematic of FRAP assay for measuring p16 dissociation-rate from tethered CDK4/6. Following photobleaching of lamin-associated mRuby3 signal, cells were monitored for recovery of mRuby3-p16 fluorescence at the nuclear periphery. **f** Quantification of the fluorescence recovery time course of mRuby3-p16, bound to CDK4- ($n = 11$ cells, 2 biological replicates) or CDK6-GFP-Δ50 lamin A ($n = 18$ cells, 2 biological replicates), after photobleaching of mRuby3. **g** Recovery timecourse of mRuby3-p16 (D84N) from CDK6-GFP-Δ50 lamin A ($n = 23$ cells, 2 biological replicates). FRAP timecourses show mean ± s.d. Source data are provided as a Source data file.

We conclude that CDKs tethered to the inner nuclear membrane via fusion with Δ50 lamin A can be used to measure the stability of complexes containing CDKs and their binding partners. Next, we sought to characterize the stability of the core G1/S-phase cyclin-CDK and cyclin-CDK-CDK inhibitor complexes that regulate CDK4/6 and CDK2 activation and the cell-cycle entry decision.

**CIP/KIP proteins stabilize cyclin-CDK complexes.** We first focused on the stability of cyclin-CDK dimers by expressing mVenus tagged D- or E-type cyclin as well as the cognate CDK2/4/6-mTurquoise-Δ50 lamin A fusion protein. Previous work has shown that tagging cyclins and CIP/KIP proteins with fluorescent proteins does not disrupt their cellular function[31–33]. Moreover, we reasoned that the combined overexpression of the constructs and co-knockdown of endogenous CIP/KIP proteins, p21 and p27, ensures that a majority of cyclin-CDK complexes do not have bound p21 or p27. The effectiveness of the co-knockdown of

p21 and p27 was validated by immunofluorescence following the FRAP experiment (Supplementary Fig. 2a). For all cyclin-CDK complexes, an enrichment could be observed of specific cyclins with their cognate CDKs at the nuclear periphery, indicating that cyclin D can bind CDK4/6 and cyclin E can bind CDK2 and form dimeric complexes in the nuclear environment (Fig. 2a, Supplementary Fig. 2b). We confirmed that formation of cyclin-CDK dimers does not require p21 or p27 by showing that cyclin D1 still bound CDK4 in p21/p27 double-knockout MEFs (Supplementary Fig. 2c).

Using FRAP to quantify dissociation times, we tested the strength of the cyclin D1-CDK4 interaction. We found that cyclin D1-CDK4 bound with a dissociation t-half of approximately 45 s (Fig. 2a, b, Supplementary Fig. 2a). A similar t-half was also found for the cyclin D1-CDK6 interaction (Fig. 2b, Supplementary Fig. 2a). Moreover, we measured the dissociation t-half between cyclin E1 and CDK2 to be 3 min, approximately four times slower than the cyclin D dissociation from CDK4/6 complexes (Fig. 2b). These results indicate that dimeric cyclin-CDK complexes,

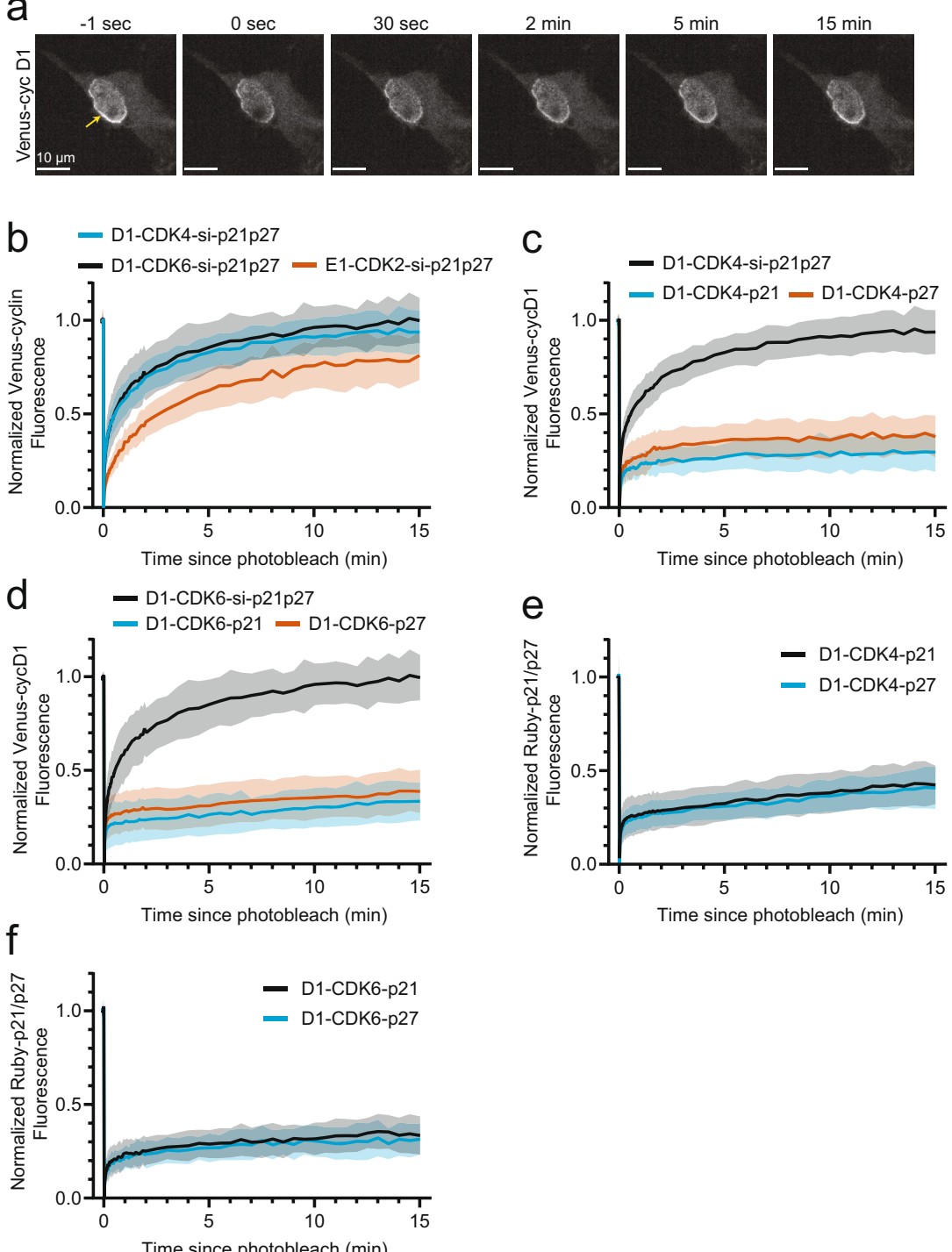

**Fig. 2 Cyclin-CDKs form a stable dimeric complex that is further stabilized by p21/p27 binding. a** Confocal fluorescence images of mVenus-cyclin D1 following photobleaching and dissociation from CDK4-mTurquoise-Δ50 lamin A complexes in cells depleted of p21 and p27 by siRNA-mediated co-knockdown. **b** Comparison of cyclin D1-CDK4 ($n = 26$ cells, 3 biological replicates), cyclin D1-CDK6 ($n = 14$ cells, 2 biological replicates) and cyclin E1-CDK2 ($n = 18$ cells, 3 biological replicates) mVenus-cyclin FRAP timecourses in cells depleted of p21 and p27. **c** mVenus-cyclin D1 FRAP timecourses in CDK4-mTurquoise-Δ50 lamin A complexes comparing cells depleted of p21 and p27 ($n = 26$, data from (**b**)) to cells co-transfected with mRuby3-p21 ($n = 14$ cells, 3 biological replicates) or mRuby3-p27 ($n = 10$ cells, 2 biological replicates). **d** mVenus-cyclin D1 FRAP timecourses in CDK6-mTurquoise-Δ50 lamin A complexes comparing cells depleted of p21 and p27 ($n = 14$; data (**b**)) to cells co-transfected with mRuby3-p21 ($n = 15$ cells, 3 biological replicates) or mRuby3-p27 ($n = 13$ cells, 2 biological replicates). **e** mRuby3-p21 ($n = 14$ cells, 3 biological replicates) or mRuby3-p27 ($n = 14$ cells, 3 biological replicates) FRAP timecourses in mVenus-cyclin D1 and CDK4-mTurquoise-Δ50 lamin A complexes. **f** mRuby3-p21 ($n = 10$ cells, 3 biological replicates) or mRuby3-p27 ($n = 11$ cells, 3 biological replicates) FRAP timecourses in mVenus-cyclin D1 and CDK6-mTurquoise-Δ50 lamin A complexes. FRAP timecourses show mean ± s.d. Source data are provided as a Source data file.

including cyclin D-CDK4/6 dimers, can form in the nucleus even under conditions in which CIP/KIP inhibitors are limiting or absent. Moreover, the cyclin D dissociation time of less than a minute argues that dimeric cyclin D-CDK4/6 complexes can exist in the nucleus but also explains that such reversibly bound dimers are difficult to isolate using immunoprecipitation due the repeated wash steps used to isolate cyclin D-CDK4/6 dimers from cellular extracts.

Having measured the dissociation times of dimeric cyclin-CDK complexes, we next investigated the effect of CIP/KIP proteins, p21 and p27, on cyclin dissociation from their cognate CDKs. To study these trimeric complexes, we also overexpressed mRuby3-tagged p21 or p27 and measured the effect of p21 or p27 on the dissociation times of cyclin D1-CDK4, cyclin D1-CDK6, and cyclin E1-CDK2. Markedly, the presence of p21 or p27 resulted in a dramatic stabilization of cyclin binding to the cognate CDK for all trimeric complexes (Fig. 2c, d and Supplementary Fig. 2d). In the 15-min time course of our FRAP assay, no recovery of the different cyclins was observed. These results are consistent with the previous biochemical work showing that the presence of p21 and p27 increases the binding affinity between cyclin D1 and CDK4[34]. We confirmed that p21 or p27 binding required cyclin-CDK dimers by transfecting CDK4 and p21 or p27 constructs into cyclin D1/D2/D3 triple-knockout MEFs. No enrichment of mRuby3-tagged p21 or p27 at the nuclear periphery was observed in the absence of cyclin D, confirming p21 and p27 only bound CDK4/6 in trimeric complexes in the nuclear setting (Supplementary Fig. 2f). mVenus-Cyclin D1 recovery modestly decreased in cells with only endogenous p21 and p27 present, likely reflecting the stabilizing effect of sub-stoichiometric levels of endogenous CIP proteins on a fraction of cyclin D1-CDK4 complexes formed using the fluorescently tagged overexpression system (Supplementary Fig. 2e).

We also used FRAP analysis to measure the dissociation of p21 and p27 from the same cyclin-CDK complexes. Like the cyclins, p21 or p27 fluorescence showed little recovery in the 15-min time course in cyclin D1-CDK4, cyclin D1-CDK6, and cyclin E1-CDK2 complexes (Fig. 2e, f and Supplementary Fig. 2g). These results demonstrate that not only cyclins but also CIP/KIP proteins are bound in CDK4/6 or CDK2 trimeric complexes with very high affinity. These results support a previously reported high-affinity binding of the p27 inhibitory domain to cyclin D1-CDK4[21].

**Clinical CDK4/6 inhibitors immediately and selectively dissociate p21 from cyclin D-CDK4 but not from cyclin D-CDK6.**
Small-molecule inhibitors of cyclin D-CDK4/6 kinase activity are approved for breast cancer therapy and are currently in numerous additional clinical trials targeting an array of cancer types. However, how these inhibitors affect CDK4/6 complexes remains poorly understood. We, therefore, used the live-cell binding system to examine whether clinical CDK4/6 inhibitors perturb CDK4/6 complex stability and potentially regulate cell-cycle entry via a non-catalytic mechanism.

We first examined how palbociclib affects the stability of nuclear cyclin D1-CDK4-p21 and cyclin D1-CDK4-p27 complexes. Following transfection, cells were incubated with palbociclib overnight and throughout the experiment. We found that cyclin D1-CDK4-p21 and cyclin D1-CDK4-p27 trimeric complexes still formed in the presence of palbociclib (Fig. 3a, Supplementary Fig. 3a). Strikingly, when we used FRAP to quantify the dissociation of p21 from cyclin D1-CDK4, we observed rapid recovery of p21 fluorescence in the presence of palbociclib (Fig. 3a, b), in stark contrast to the very slow p21 recovery observed in the absence of palbociclib (Figs. 3a, 2e). The

accelerated dissociation rate of p21 from the cyclin D1-CDK4-p21 complexes suggests that palbociclib greatly decreases the affinity of p21 in the cyclin D1-CDK4-p21 complex. In contrast, we found no differences in the slow p27 fluorescence recovery in trimeric cyclin D1-CDK4-p27 complexes in the presence or absence of palbociclib (Figs. 3b, 2e). Moreover, we found that cyclin D1-CDK4 complexes also still formed following co-knockdown of p21/p27 and overnight treatment with palbociclib, demonstrating that cyclin D-CDK4 dimers form in the nucleus both in the presence or absence of clinical CDK4/6 inhibitors (Supplementary Fig. 3b).

To analyze the effect of palbociclib on already formed cyclin D1-CDK4-p21 complexes, we acutely added palbociclib after a delay following p21 photobleaching. Between photobleaching and the delayed palbociclib addition, there was minimal p21 recovery (Fig. 3c and Movie S2), consistent with our prior results (Fig. 2e). Markedly, immediately upon palbociclib addition, we observed a rapid and complete recovery of the p21 fluorescence signal. Similarly, rapid recoveries were observed with the two other clinically used CDK4/6 inhibitors, ribociclib and abemaciclib (Fig. 3c). These results were validated in vitro by using complexes isolated by immunoprecipitation of CDK4, in which acute addition of palbociclib dramatically reduced the amount of bound p21 within an hour (Fig. 3d). In the reciprocal experiments, we observed decreased CDK4 levels upon acute addition of palbocicilb to complexes isolated by immunoprecipitation of p21 (Fig. 3e). Furthermore, cyclin D2-CDK4-p21 complexes responded similarly to palbociclib in the FRAP assay (Supplementary Fig. 3c), demonstrating that this effect is not limited to cyclin D1. Clinical CDK4/6 inhibitors can, therefore, rapidly access pre-formed trimeric CDK complexes. In principal, the stability of a trimeric CDK4 complex could be reduced by accelerating the dissociation rate of cyclin D, p21, or both from the trimeric complex. However, palbociclib did not change the stability of cyclin D1 in the cyclin D1-CDK4-p21 complex despite the accelerated p21 dissociation-rate (Supplementary Fig. 3d). This result demonstrates that high affinity binding of cyclin D1 to CDK4 can be maintained in the presence or absence of palbociclib by low- or high-affinity binding of p21 to cyclin D1-CDK4, respectively. We conclude that clinical CDK4/6 inhibitors act immediately, lowering the affinity of p21 by increasing the dissociation rate of p21 more than 20-fold. This effect is selective for trimeric CDK4 complexes containing p21 but not p27. Moreover, CDK4/6 inhibitors do not prevent the formation of trimeric complexes.

We next tested whether CDK4/6 inhibitors affect other CDK complexes. Acute addition of palbociclib, ribociclib, or abemaciclib did not change the p27 recovery time in cyclin D1-CDK4-p27 complexes (Supplementary Fig. 3e), consistent with the overnight incubation results (Fig. 3b). To test if CIP/KIP proteins were similarly affected in CDK6 complexes, we added CDK4/6 inhibitors acutely to cyclin D1-CDK6-p21 and cyclin D1-CDK6-p27 complexes (Fig. 3f and Supplementary Fig. 3f). Most surprisingly, no change in either p21 or p27 recovery was observed in complexes containing CDK6. We conclude that clinical CDK4/6 inhibitors are selective in two ways: they only lower p21 but not p27 affinity in trimeric complexes, and they only lower p21 affinity in complexes containing CDK4 but not CDK6.

Given the surprising difference in palbociclib-regulated p21 affinity in CDK4 versus CDK6 complexes, we investigated whether it is the N- or C-terminal lobe of the two kinases that explains the differential response to palbociclib. We first compared the binding of a minimal region of p21 (p21KID; amino acids 14-81) that can inhibit cyclin D1-CDK4 or cyclin D1-CDK6[17,35,36]. In the recently reported structure of trimeric

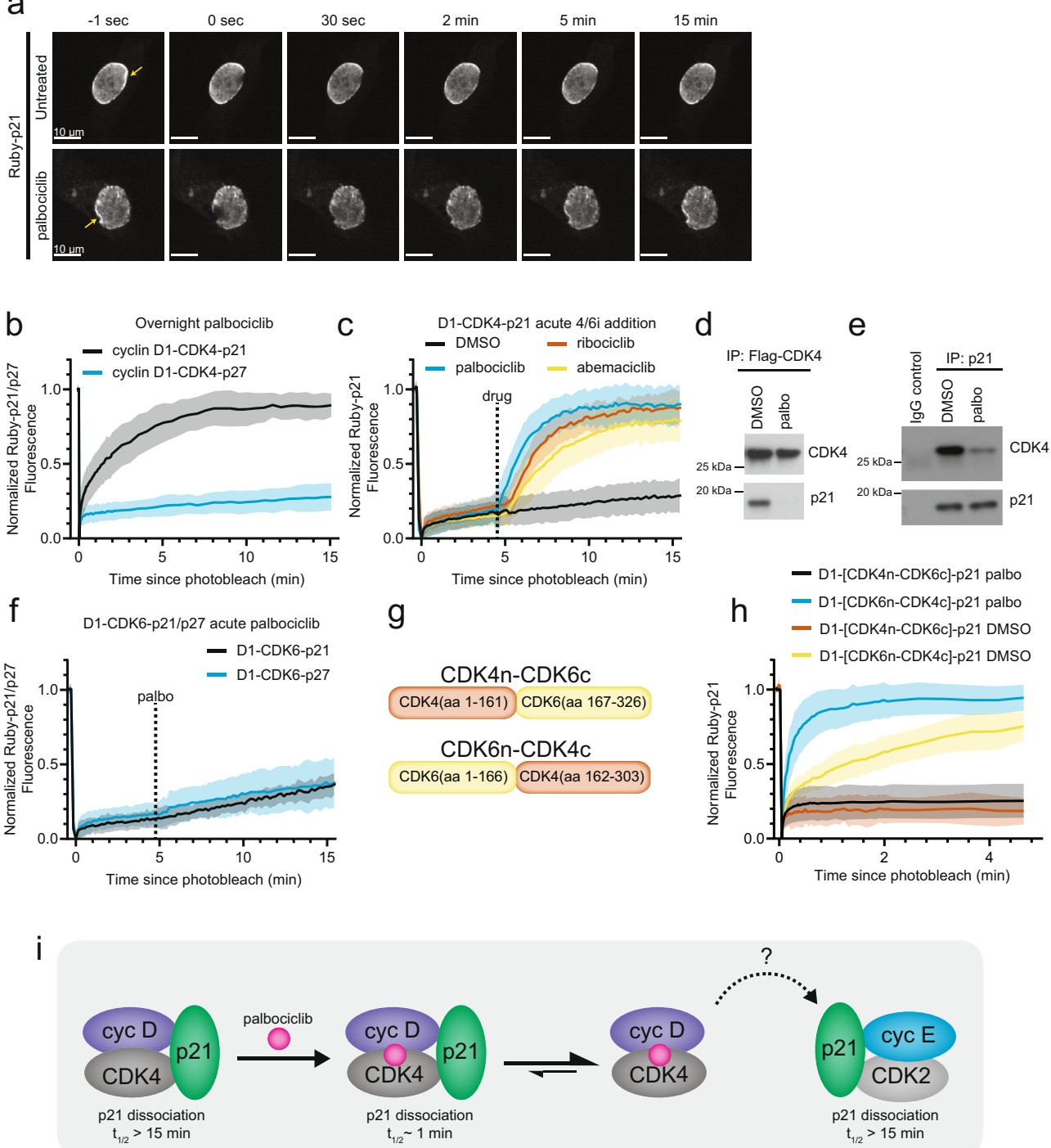

complexes, this short p21KID construct directly interacts with both cyclin D1 and the N-lobe of CDK4 but not the C-lobe of CDK4 (Supplementary Fig. 3g)[17]. We found that the p21KID behaves similarly to full-length p21: The affinity of the p21KID dramatically decreases only toward cyclin D1-CDK4 complexes in the presence of palbociclib compared to DMSO treatment, but remains tightly bound to cyclin D1-CDK6 complexes (Supplementary Fig. 3h). These results indicate that palbociclib binding acts to change the affinity of p21 predominantly through changing the interaction between p21 and the N-lobe of CDK4. Thus, we hypothesized that sequence differences between the

CDK4 and CDK6 N-lobe distinguished the response to palbociclib.

To directly test this hypothesis, we generated chimeric protein constructs containing the N-lobe of CDK4 and C-lobe of CDK6 and vice versa (Fig. 3g). Surprisingly, we found palbociclib sensitivity only with the CDK4 C-lobe, which does not directly interact with p21KID, indicating that the C-lobe is responsible for the decreased binding affinity of p21 (Fig. 3h). We note that the binding affinity of p21 to the CDK6n-CDK4c chimera in the presence of DMSO is lower than its affinity to wild type CDK4 or CDK6 in the absence of palbociclib, but the off-rate is still

**Fig. 3 Palbociclib acutely and selectively dissociates p21 from cyclin D1-CDK4-p21 complexes. a** Confocal fluorescence images comparing the recovery timecourse of mRuby3-p21 bound to cyclin D1-CDK4 following photobleaching in either the untreated condition or following overnight incubation with palbociclib. **b** FRAP timecourses of mRuby3-p21 ($n = 13$ cells, 2 biological replicates) or mRuby3-p27 ($n = 14$ cells, 3 biological replicates) in cyclin D1-CDK4-p21/p27 complexes in cells that were maintained in 6 µM palbociclib overnight and throughout the photobleaching experiment. **c** mRuby3-p21 fluorescence recovery in cyclin D1-CDK4-p21 complexes during a control addition of DMSO ($n = 14$ cells, 2 biological replicates) or acute addition of the CDK4/6 inhibitors palbociclib (6 µM; $n = 21$ cells, 3 biological replicates), ribociclib (6 µM; $n = 16$ cells, 3 biological replicates), or abemaciclib (6 µM; $n = 14$ cells, 3 biological replicates). Photobleaching was performed and recovery of the mRuby3-p21 fluorescence signal was measured. Only after a delay, at the indicated timepoint, were CDK4/6 inhibitors or DMSO acutely added while continuing the measurement of mRuby3-p21 fluorescence recovery. **d** MCF-10A cells stably expressing 3xFLAG-CDK4 were synchronized by serum starvation then lysed 14 h following mitogen stimulation. 3xFLAG-CDK4 complexes were immunoprecipitated and treated with either DMSO or palbociclib (6 µM) for 1 h, followed by immunoblotting to determine the amount of p21 bound to CDK4 complexes. **e** Experimental setup as in (**d**), but p21 complexes were immunoprecipitated and the amount of CDK4 bound to p21 was determined following 1 h treatment with palbociclib (6 µM) or DMSO. **f** mRuby3-p21 ($n = 10$ cells, 2 biological replicates) or mRuby3-p27 ($n = 10$ cells, 3 biological replicates) fluorescence recovery in cyclin D1-CDK6-p21/p27 complexes during acute addition of 6 µM palbociclib as in (**c**). **g** Schematic of CDK4-CDK6 chimeras fusing the CDK4 N-lobe to the CDK6 C-lobe or vice versa. **h** FRAP timecourses of mRuby3-p21 in cyclin D1-CDK4/6 chimera complexes in cells that were maintained in 6 µM palbociclib or DMSO overnight and throughout the photobleaching experiment ($n = 15$, 2 biological replicates CDK4n-CDK6c palbociclib; $n = 24$, 2 biological replicates CDK6n-CDK4c palbocicilib, $n = 10$, 2 biological replicates CDK4n-CDK6c DMSO, $n = 14$, 2 biological replicates CDK6n-CDK4c DMSO). **i** Model for how palbociclib may inhibit CDK2 activity by a non-catalytic mechanism. In the absence of palbociclib, p21 is tightly bound in a trimeric cyclin D-CDK4-p21 complex and dissociates very slowly. Treatment with palbociclib immediately interacts with cyclin D-CDK4-p21 trimeric complexes and greatly increases the dissociation rate and reduces the affinity of p21 from cyclin D-CDK4. The high binding affinity of p21 toward CDK2 complexes, which is unaffected by palbociclib, is then expected to cause a redistribution of p21 from CDK4 to CDK2 complexes, resulting in a non-catalytic inhibition of CDK2 activity. FRAP curves show mean ± s.d. palbo, palbociclib. Source data are provided as a Source data file.

reduced in the presence of palbociclib by approximately 10-fold (Fig. 3h and Fig. 2e, f). This result suggests that palbociclib binding generates an allosteric structural change in cyclin D-CDK4 that lowers p21 affinity, and this structural change is enabled by unique sequences in the C-lobe of CDK4. This mechanism allows the common p21 binding site to inhibit both CDK4 and CDK6 for normal function, and, in regulation that is unique to CDK4 and p21, allows binding of palbociclib to control p21 affinity through differential interactions at an allosteric site. Similar allosteric regulation of binding interactions has been reported to differentially regulate closely homologous proteins in the diversification of MAP and SRC family kinases[37,38].

In control experiments, the slow dissociation of p21 from cyclin E1-CDK2-p21 complexes was also unaffected by addition of different CDK4/6 inhibitors, including abemaciclib (Supplementary Fig. 3i), which also inhibits CDK2 at high concentrations[13]. CDK2 trimeric complexes are therefore generally unaffected by clinical CDK4/6 inhibitors. Thus, as a consequence of the lower affinity and rapidly reversible binding of p21 to cyclin D-CDK4 in the presence of palbociclib, one could expect that in cells expressing p21 and CDK4, palbociclib treatment may result in a redistribution of p21 from cyclin D-CDK4 to cyclin E-CDK2, since CDK2 complexes can still bind p21 with high affinity (Fig. 3i).

**Palbociclib inhibits CDK4/6 catalytically and CDK2 non-catalytically to synergistically inhibit cell-cycle entry.** To test the hypothesis that clinical CDK4/6 inhibitors can have a rapid non-catalytic role, we first synchronized MCF-10A cells by serum starvation and mitogen release, then treated cells with either palbociclib or DMSO for 30 min before we immunoprecipitated with antibodies targeting endogenous p21. Consistent with the hypothesis, treatment of only 30 min with palbociclib robustly increased the amount of CDK2 and decreased the amount of CDK4 when we pulled down p21 (Fig. 4a). We also tested this hypothesis in two estrogen receptor-positive breast cancer lines, MCF7 and T-47D, which may better model breast cancers treated by palbociclib in the clinic. In both cell lines we found a robust decrease in CDK4 levels pulled down by p21 in the presence of palbociclib (Supplementary Fig. 4a, b). We found an increase in CDK2 pull down in the presence of palbociclib in T-47Ds, but not

MCF7s. Given the non-catalytic effect of palbociclib is specific to CDK4 complexes, the extent of p21 redistribution could be dependent on the relative concentrations of CDK4, CDK6, CDK2, and p21. Together with the FRAP analysis, this biochemical analysis argues that palbociclib treatment can cause a redistribution of p21 from CDK4 complexes to CDK2 complexes.

To determine whether redistribution of p21 affects the activity of CDK2, we used a similar experimental design but now combined live-cell imaging of a fluorescent reporter that measures the activity of cyclin E-bound CDK2 in G1 phase with computational single-cell tracking to quantify CDK2 activity in individual cells over time (Fig. 4b)[39–42]. To analyze a potential effect of palbociclib-induced p21 release from cyclin D-CDK4-p21 on inhibiting CDK2 activity, we computationally gated for cells with active CDK2 and excluded cells that had entered S phase, since p21 is rapidly degraded when cells enter S phase[43–47]. Examination of the CDK2 median activity traces showed a rapid decrease in CDK2 activity following palbociclib addition (Fig. 4c), consistent with the previous reports[18]. However, cells pre-treated with siRNA targeting p21 did not exhibit a similarly rapid decrease in CDK2 activity (Fig. 4c, d). This suggests that redistribution of p21 protein away from cyclin D-CDK4-p21 mediates the sudden decrease in CDK2 activity following palbociclib addition, since cells that lack p21 mostly maintain their CDK2 activity at an intermediate level. Consistent with the results of our FRAP assay, siRNA knockdown of p27 protein did not phenocopy the effect of the p21 knockdown (Fig. 4c, d). We observed similar results upon ribociclib addition, indicating that the p21-dependent acute inhibition of CDK2 activity is not restricted to palbociclib (Supplementary Fig. 4d, e). All siRNA knockdowns were confirmed by immunofluorescence (Supplementary Fig. 4c). We conclude that palbociclib has an immediate non-catalytic role in cells to redistribute p21 away from cyclin D-CDK4-p21 to bind CDK2 complexes and inhibit CDK2 activity.

Finally, we tested whether p21 levels have an effect on the rapid palbociclib-mediated loss of Rb hyper-phosphorylation in G1 that has been previously reported[18]. To test for an effect of p21, we treated mitogen-stimulated cells containing the CDK2 activity sensor with palbociclib as above but instead fixed cells within 15 min following palbociclib addition (Fig. 4e). We computationally gated for cells that were in G1 with low CDK2 activity and

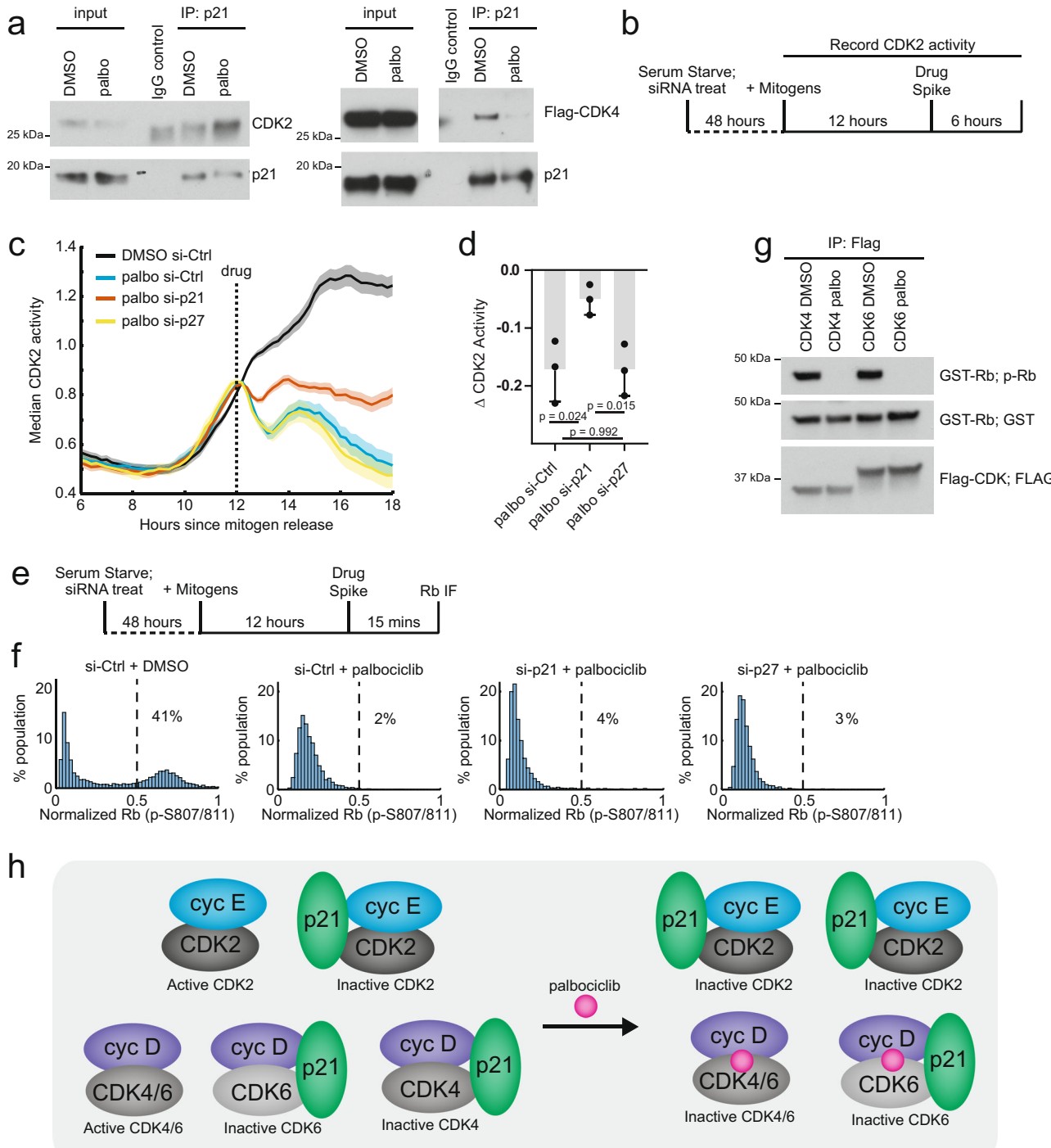

analyzed whether cells maintained hyper-phosphorylated Rb. In control siRNA-treated cells, palbociclib addition effectively dephosphorylated Rb in less than 15 min (Fig. 4f). This result demonstrates that active CDK4/6 complexes are immediately accessible to and inhibited by palbociclib. siRNA knockdown of p21 did not rescue this effect, as cells still rapidly dephosphorylated Rb following palbociclib treatment even in the absence of p21. Similar results were observed with siRNA knockdown of p27 and treatments with ribociclib (Supplementary Fig. 4f). A direct inhibition of CDK4/6 activity was further supported by in vitro kinase assays of immunoprecipitated 3xFLAG-CDK4 and 3xFLAG-CDK6 complexes, where the activity of both CDK4 and CDK6 complexes toward Rb was

rapidly inhibited by addition of palbociclib, contrasting with a recent report of palbociclib mechanism of action (Fig. 4g)[17]. We conclude that clinical CDK4/6 inhibitors have a catalytic role in cells to rapidly inhibit CDK4/6 kinase activity towards Rb as well as a selective non-catalytic role to rapidly redistribute p21 away from CDK4 to inhibit CDK2 activity.

## Discussion

We developed a method to measure the stability of nuclear protein complexes in live cells, allowing us to evaluate how clinical CDK4/6 inhibitors regulate CDK complexes. Our results show that dimeric cyclin D1-CDK4 complexes, which have a dissociation time of 45 s, can form in the nucleus. Additionally,

**Fig. 4 Palbociclib inhibits CDK2 via a non-catalytic mechanism by redistribution of p21 from cyclin D-CDK4. a** MCF-10A cells stably expressing 3xFLAG-CDK4 where synchronized by serum starvation then stimulated with mitogens for 14 h prior to 30-min treatment with DMSO or palbociclib (6 µM). Immunoprecipitation of p21 was then performed and the amount of p21-bound CDK4 or CDK2 was determined by immunoblotting. **b** Schematic of the experimental protocol used to measure changes in CDK2 activity following palbociclib addition. Cells were serum starved and treated with the indicated siRNA immediately after initiating serum starvation and maintained in starvation media for 48 h. Cells were then stimulated with mitogens and live-cell imaging of CDK2 activity was performed. At the 12-h timepoint, DMSO or palbociclib was acutely added and imaging was continued. **c** Median live-cell CDK2 activity traces in mitogen-stimulated MCF-10A cells treated with indicated siRNA and DMSO or palbociclib (3 µM). Cells were selected for the analysis if they had CDK2 activity levels between 0.65 and 0.75 at the time of drug addition and if they had not yet entered S phase, as determined by FUCCI-APC/C$^{CDH1}$ fluorescent reporter. Shaded areas indicate 95% confidence intervals. Data are from one experiment representative of three biological replicates. (DMSO si-Ctrl $n = 281$ cells, palbo si-Ctrl $n = 173$ cells, palbo si-p21 $n = 207$ cells, palbo si-p27 $n = 103$ cells). **d** Bar graph showing average drop in CDK2 activity following addition of palbociclib in si-Ctrl, si-p21, and si-p27 cells, comparing the difference in CDK2 activity at the time of drug spike and the average activity in the last hour of the time course. The results of three biological replicates are shown for each condition (mean ± s.d). One of the replicates is from data shown in (**c**). Replicate 1: DMSO si-Ctrl $n = 281$ cells, palbo si-Ctrl $n = 173$ cells, palbo si-p21 $n = 207$ cells, palbo si-p27 $n = 103$ cells; Replicate 2: DMSO si-Ctrl $n = 119$ cells, palbo si-Ctrl $n = 124$ cells, palbo si-p21 $n = 156$ cells, palbo si-p27 $n = 174$ cells Replicate 3: DMSO si-Ctrl $n = 99$ cells, palbo si-Ctrl $n = 47$ cells, palbo si-p21 $n = 546$ cells, palbo si-p27 $n = 73$ cells. Statistical analysis via a two-tailed unpaired Student's $t$-test. **e** Experimental schematic for measuring acute loss of Rb hyper-phosphorylation upon palbociclib addition. Cells prepared as in (**b**), but cells were fixed 15 min after drug addition and Rb hyper-phosphorylation analyzed by immunofluorescence. **f** Histogram of Rb (phospho-S807/811) over the total Rb (phosphosite-independent antibody) immunofluorescence. Cells were gated for those with CDK2 activity <0.6 and no S-phase entry, as determined by FUCCI APC/C$^{CDH1}$ fluorescent reporter. For each cell, nuclear intensity of p-S807/811 Rb and total Rb was measured and a ratio of the intensities was calculated. Cells were then normalized to the maximum single-cell ratio and plotted. Percentage of cells with a ratio greater than 0.5, representing cells with hyper-phosphorylated Rb, was calculated for each condition. DMSO si-Ctrl $n = 5204$ cells, palbo si-Ctrl $n = 4252$ cells, palbo si-p21 $n = 4176$ cells, palbo si-p27 $n = 4345$ cells. Plotted data are representative of three biological replicates. **g** IP kinase assay of immunoprecipitated 3xFLAG-CDK4 or 3xFLAG-CDK6 complexes towards GST-Rb (C terminus). CDK4 or CDK6-mediated phosphorylation of Rb (807/811) was assessed in the presence of DMSO control or palbociclib (6 µM) for 30 min. **h** Model of the catalytic and non-catalytic mechanisms how clinical CDK4/6 inhibitors inhibit CDK4/6 and CDK2 activity, respectively. In palbociclib treated conditions, active CDK4/6 complexes are catalytically inhibited by binding palbociclib while cyclin E-CDK2 complexes are non-catalytically inhibited by the transfer of p21 from cyclin D-CDK4-p21 to cyclin E-CDK2 as a result of the palbociclib mediated selective destabilization of cyclin D-CDK4-p21. Source data are provided as a Source data file.

these dimeric cyclin D1-CDK4 complexes, as well as the much higher affinity trimeric cyclin D1-CDK4-p21/p27 complexes, still form in the presence of palbociclib. Strikingly, treatment with clinical CDK4/6 inhibitors immediately destabilizes the binding of p21 in pre-formed trimeric cyclin D-CDK4-p21 complexes. The destabilization is restricted to complexes that contain p21, since CDK4 trimeric complexes containing p27 were unaffected by clinical CDK4/6 inhibitors. Surprisingly, the destabilization is also restricted to CDK4 kinase, since trimeric complexes containing CDK6 kinase and either p21 or p27 were also unaffected by clinical CDK4/6 inhibitors.

Our findings contrast with a recent report suggesting that palbociclib predominantly binds monomeric CDK4 and cannot access pre-formed CDK complexes[17]. Based on the proposed inaccessibility of pre-formed CDK4 complexes to palbociclib, the authors further suggested the cell-cycle inhibitory effect of palbociclib occurred on the time scale of days by suppressing an early folding/assembly step. In contrast, we observe that dimeric cyclin D-CDK4/6 and trimeric CDK4/6 complexes containing p21 or p27 still form in the nucleus in the continued presence of clinical CDK4/6 inhibitors. Moreover, we demonstrate two inhibitory mechanisms occurring within minutes following addition of clinical CDK4/6 inhibitors—a direct p21-independent inhibition of CDK4/6 activity and an indirect p21-mediated inhibition of CDK2 activity (Fig. 4h). We conclude that clinical CDK4/6 inhibitors could have two roles in cells: a catalytic role to inhibit cyclin D-CDK4/6 activity and an unexpected, rapid non-catalytic role to inhibit CDK2 activity through a selective redistribution of p21 away from trimeric cyclin D-CDK4-p21.

Our discovery of a new mechanism of palbociclib inhibition and its potential implications described below should motivate future testing in clinical settings. While inhibition of both CDK4 and CDK2 activities contribute to the suppression of cell-cycle entry, the restricted non-catalytic effect only for cyclin D-CDK4-p21 trimers may reduce the efficacy of and facilitate resistance to current clinical CDK4/6 inhibitors. Our results show that in the absence of p21,

CDK2 activity remains elevated during G1 following palbociclib treatment, while CDK2 activity can be partially or fully suppressed in the presence of p21. This suggests that the non-catalytic palbociclib-mediated redistribution of p21 to CDK2 complexes increases the barrier for cell-cycle entry by suppressing the ability of cells to further increase CDK2 activity and enter the cell cycle. Consistent with this model, increased CDK2 activity, e.g., via cyclin E overexpression, is a known palbociclib resistance mechanism that may overcome the restricted non-catalytic effect[48–51]. The CDK4-specific effect of palbociclib on displacing p21 from cyclin D1-CDK4-p21—but not from cyclin D1-CDK6-p21—suggests that the relative ratio of CDK4 to CDK6 within a tumor may dictate drug sensitivity and resistance. Elevated expression of CDK6 could bind and sequester p21, preventing p21-mediated CDK2 inhibition in the presence of palbociclib. Indeed, prior studies have found that CDK6 overexpression commonly occurs during tumorigenesis and increases the resistance to CDK4/6 inhibitors in cancer cell lines[6,13–16]. Our studies using CDK4 and CDK6 chimeras and a truncated p21 inhibitory protein unexpectedly showed that it is the C-lobe of CDK4 that selectively controls a palbociclib-mediated reduction in p21 affinity, arguing for an allosteric selectivity mechanism. While abemaciclib and ribociclib displaced p21 from CDK4 complexes akin to palbociclib, further work should investigate whether these effects are synonymous in the clinic. Given the common selectivity amongst the current generation of CDK4/6 inhibitors in vitro, improved CDK4/6 inhibitors may be needed. A second generation of CDK4/6 inhibitors with a broadened non-catalytic mechanism that also destabilizes cyclin D-CDK6-p21 complexes and complexes containing p27 may overcome resistance mechanisms and improve the efficacy of CDK4/6 inhibitors as anticancer therapeutics.

## Methods

**Cell culture**. hTERT RPE-1 cells (ATCC, #CRL-4000) were maintained in phenol red-free DMEM/F12 supplemented with 10% FBS. MCF-10A cells (ATCC, #CRL-10317, RRID:CVCL_0598, human female) were maintained in growth media

consisting of phenol red-free DMEM/F12 supplemented with 5% Horse Serum, 20 ng/mL EGF, 10 μg/mL insulin, 500 μg/mL hydrocortisone, and 100 ng/mL cholera toxin. MCF-10A starvation media consisted of phenol red-free DMEM/F12 supplemented with 500 μg/mL hydrocortisone, 100 ng/mL cholera toxin, and 0.3% BSA. During passaging, 0.05% trypsin-EDTA was inactivated with phenol red-free DMEM/F12 supplemented with 20% horse serum. Immortalized mouse embryonic fibroblast (MEF) lines were cultured in DMEM supplemented with 10% FBS and 50 U/mL penicillin and 50 μg/mL streptomycin (1% P/S). MCF7 cells were cultured in Eagle's Minimum Essential Media supplemented with 0.01 mg/mL human recombinant insulin and 10% fetal bovine serum. T-47D cells were cultured in RPMI-1640 Medium supplemented with 0.01 mg/mL human recombinant insulin and 10% fetal bovine serum.

**DNA constructs and cell lines**. CDK4- and CDK6-EGFP-Δ50 lamin A were generated by first PCR amplifying EGFP-Δ50 lamin A from the plasmid pEGFP-Δ50 lamin A, which was a gift from Tom Misteli (Addgene plasmid # 17653; http://n2t.net/addgene:17653; RRID:Addgene_17653). PCR products for Δ50 lamin A and CDK4 or CDK6 were combined via Gibson Assembly into a C1 backbone (Clonetech). All cyclin, CIP/KIP, and INK4 transfection constructs were generated via Gibson assembly using a C1 backbone (Clonetech). mRuby3-p16 was made using a custom p16 codon-optimized sequence. A list of primers used to generate these constructs can be found in Supplementary Table 1.

MCF-10A cells for CDK2 activity experiments were generated via transduction with lentiviral vectors encoding CSII-pEF1a-H2B-mTurquoise, CSII-pEF1a-DHB (aa994-1087)-mVenus, and CSII-pEF1a-mCherry-Geminin(aa1-110) as described previously[42,52]. Following transduction, cell sorting for this project was done on a Becton Dickinson Influxin at the Stanford Shared FACS to obtain populations expressing the fluorescent reporters. MCF-10A cells expressing 3xFlag-CDK4 were generated via transduction with lentiviral vectors encoding pLV-EF1a-3xFlag-CDK4-IRES-puro and puromycin selection.

**FRAP experimental setup**. A step-by-step protocol describing the FRAP assay can be found at Protocol Exchange[53]. On day 1, a 96-well plate (Cellvis) was coated with 0.3 mg/ml collagen (Advanced BioMatrix, 5005-B) in PBS for 3 h at 37 °C. RPE-1 or cells were then resuspended and seeded onto a glass-bottom at 6000 cells/well to achieve 50–70% confluency after 48 h. On day 2, cells were transiently transfected with 0.25–0.5 μg/well of indicated plasmids resuspended in 100 μL OptiMEM (ThermoFisher) plus 0.25 μL Lipofectamine 2000 (ThermoFisher) and incubated for 3 h at 37 °C prior to washing and overnight incubation at 37 °C in maintenance media. For conditions with overnight treatment with palbociclib, the transfection media was replaced with fresh maintenance media containing palbociclib (6 μM). For experiments entailing siRNA co-knockdown of p21 and p27, cells were first transfected with siRNA resuspended to a final concentration of 20 nM in 100 μL OptiMEM plus 0.2 μL DharmFECT 1 (Dharmacon) and incubated at 37 °C followed by washout of siRNA transfection media and a 1-h rest in maintenance media before proceeding to plasmid transfection. Non-targeting control siRNA (D-001810-10-05) and siRNA targeting CDKN1B (p27) (J-003472-07) or CDKN1A (p21) (M-003471-00) were obtained from Dharmacon.

On day 3, 100 μL fresh maintenance media was added and cells were rested for at least 1 h prior to proceeding to imaging. Imaging was performed in an incubated chamber maintained at 37 °C. Prior to FRAP, we treated cells with a cocktail of drug treatments to prevent cellular movement. First, 50 μL of maintenance media plus Y-27632 was added to a final concentration of 40 μM Y-27632 and incubated for 20 min. An additional 50 μL of maintenance media plus drugs was then added to achieve the following concentrations: Y-27632 (40 μM; Cell Signaling Technologies), Jasplakinolide (8 μM; Cayman Chemical), Latrunculin B (10 μM; Abcam), and nocodazole (1 μg/ml; Sigma-Aldrich). Following a 30-min incubation, FRAP-based imaging was then performed on cells that expressed all the constructs of interest and had sufficient expression of the CDK-Δ50 lamin A fluorescent construct that a ring of localization could be observed for the interacting protein of interest. The phase of the cell-cycle was not known for these experiments. Four to five cells were imaged per well after cytoskeleton stabilization and no analysis was initiated more than 1 h after cytoskeleton stabilization. For acute treatments, an additional 50 μL of RPE-1 media containing DMSO or clinical CDK4/6 inhibitors and maintaining the final concentration of cytoskeleton stabilization drugs was added to the wells approximately 4–5 min after photobleaching. FRAP images were acquired using a fully automated widefield/Yokogawa spinning-disc confocal fluorescence microscope system (Intelligent Imaging Innovations, 3i) using a 60× 1.27 NA water-immersion objective. The system was built around a Nikon Ti-E stand and utilized a 3i laser stack (405, 445, 488, 514, 561, 640 nm), a 3i Vector photomanipulation device, a Yokogawa CSU-W1 scanning head with dual camera port, two sCMOS cameras (Andor Zyla 4.2), enclosed by an environmental chamber (Haison), and controlled by SlideBook software (3i).

MEF transfection was performed according to the above protocol and cells were imaged on day 3 without being treated with the cytoskeleton freezing drug cocktail.

**FRAP analysis**. Fluorescence intensity was measured using Slidebook 6.13 software. FRAP curves were generated from raw intensity values according to Equation 1.

$$\text{CORRECTED CURVE} = (\text{ROI} - \text{BG})/(\text{REF} - \text{BG}) \qquad (1)$$

Where the region of interest (ROI) is the area of photobleaching at the nuclear lamin, reference region (REF) is a distant nucleoplasmic region within the cell, and background (BG) is a nearby region absent any cells. FRAP curves were then normalized from 0 to 1 by subtracting the first post-photobleaching value from all timepoints, then dividing all timepoints by the pre-photobleaching corrected values. Analysis was performed using Excel365 and Graphpad Prism v8.0. All FRAP curves were generated using cells acquired on at least two separate days of experimental setup.

**IP and IP-Kinase experiments**. For in vitro palbociclib addition experiments, MCF-10A cells expressing 3xFLAG-CDK4 were plated in two 10 cm diameter cell culture dishes and grown for two days before being switched into starvation media for two days. Cells were released for 14 h and harvested by washing once with cold PBS prior to the addition of 250 μL lysis buffer (50 mM HEPES pH = 7.5, 150 mM NaCl, 1 mM EDTA, 2.5 mM EGTA, 10% glycerol, 0.1% Tween-20), containing PhosSTOP (Sigma-Aldrich 4906837001), 1 mM DTT, and EDTA-free Halt Protease Inhibitor Cocktail (Thermo Fisher Scientific 78439). Cells were lysed for 30 min on ice with occasional vortexing, then centrifuged at $10,000 \times g$ for 10 min at 4 °C. Supernatants were combined and approximately 0.6 mg of lysate was incubated with anti-FLAG magnetic beads (Sigma-Aldrich M8823) for 30 min at 4 °C. Following incubation, the IP was split into two equal conditions of approximately 0.25 mg of sample and treated with either DMSO or palbocicilb (6 μM final). Samples were incubated at 37 °C for one hr with mixing, then beads were washed 3× with lysis buffer. Samples were eluted off the beads with 6× Laemmli buffer (0.375 M Tris pH = 6.8, 12% SDS, 60% glycerol, 0.6 M DTT, 0.06% Bromophenol blue) and boiling. Western blots were performed using α-CDK4 (Abcam ab108357, 1:2000) and α-p21 (CST 2947 S, 1:1000). Detection was performed using HRP-conjugated anti-mouse (CST 7076, 1:5000) or anti-rabbit (CST 7074, 1:5000) secondary antibodies and chemiluminescent substrate (Thermo Scientific 34080). For all Westerns, IP inputs are from the same blots as experimental samples, and the IP bait controls are always from the same lane on the same blot as the experimental prey analysis. All uncropped blots can be found in the Source data file.

A similar procedure was performed in wild type MCF-10A cells for the reciprocal experiment in which p21 was immunoprecipitated with a few differences. Samples were divided into 0.25 mg of sample for a rabbit IgG control (CST 2729) and 0.6 mg was used for rabbit p21 (CST 2947S) IP. IPs were incubated at 4 °C for 30 min and captured by addition of Protein G beads (Thermo Fisher 10004D, 35 μL per 0.25 mg of sample). Following capture, the p21 IP was split into two equal conditions of 0.25 mg of sample and treated with a either DMSO or palbocicilb (6 μM final) for 1 h at room temperature with mixing. Beads washing and elution was performed as above. Samples were evaluated by Western blot, using α-CDK4 (Santa Cruz sc-56277, 1:500) and α-p21 (Fisher SX118, 1:500).

For cellular treatment experiments, MCF-10A cells expressing 3xFLAG-CDK4 were plated in three 10 cm diameter cell culture dishes and grown for 2 days before being switched into starvation media for two days. Cells were released in growth media for 14 h and treated with a spike of either DMSO or palbociclib (6 μM final) for 30 min at 37 °C before being lysed as above, but in the continued presence of DMSO or palbociclib. Following lysis and centrifugation, ~0.25 mg of lysate was incubated with either 0.5 μg of rabbit IgG control or rabbit p21 antibody as described above. Following capture with Protein G beads, beads were washed in lysis buffer containing DMSO or palbociclib and eluted as above. For MCF7 and T-47D experiments, cycling cells were used for the IP protocol outlined above. IPs were evaluated by western blot using α-p21, α-FLAG (Sigma F3165, 1:1000), and α-CDK2 (Origene OTI2A5, 1:2000).

For IP-kinase assays, cycling MCF-10A cells expressing either 3xFLAG-CDK4 or 3xFLAG-CDK6 were plated in a 10 cm dish and lysed as described above. IPs were performed by incubation with anti-FLAG magnetic beads (Sigma-Aldrich M8823) for 30 min at 4 °C. Following incubation, the beads were washed 3× with lysis buffer and 2× with 50 mM HEPES pH = 7.5 and 1 mM DTT. For the kinase assay, samples were split and beads were resuspended in 30 μL kinase reaction buffer (50 mM HEPES pH = 7.5, 10 mM MgCl₂, 2.5 mM EGTA, 1 mM DTT, 50 μM ATP (Cytoskeleton #BSA04), 1 μg GST-Rb C-terminus (sc-4112)) containing either DMSO or palbociclib, and incubated at 37 °C for 30 min with continuous agitation. Kinase reactions were denatured by adding 6x Laemmli buffer and boiling. Kinase activity was determined by western blot, iteratively blotting, using α-GST (Santa Cruz sc-138, 1:1000), α-Rb (p-807/811) (CST 8516, 1:1000), and α-FLAG (Sigma F3165, 1:1000).

**Live-cell CDK2 reporter experiments**. MCF-10A cells were seeded into a 96-well plate (Costar #3904) at a density of 10,000 cells per well ~24 h prior to quiescence induction. To induce quiescence, cells were washed with PBS then maintained in starvation media (described above) for 48 h prior to mitogen release. One hour following the starvation of cells, control, p21-targeting, or p27-targeting siRNA were transfected into cells at a final concentration of 20 nM per well with 0.2 μL Dharmafect 1. After 6 h of incubation, cells were transferred back into starvation media. Cells were released from starvation with 200 μL fresh growth media and image acquisition was initiated. Imaging was performed in an incubated chamber maintained at 37 °C and 5% CO₂ using an automated fluorescence microscope (IXμ, Molecular Devices, MetaExpress version 6.1) with a 10× 0.3NA objective.

Images were acquired in the CFP, YFP, and Texas Red channels every 12 min with an overall exposure time under 600 ms.

**Inhibitors**. Palbociclib (PD-0332991) (Selleck Chem S1116), Ribociclib (LEE011) (Selleck Chem S7440), and Abemaciclib (LY2835219) (Selleck Chem S7158) were used at a concentration of 6 μM for FRAP experiments. Palbociclib and Ribociclib were used at a concentration of 3 μM for live-cell CDK2 activity experiments in MCF-10A cells.

**Image analysis**. Cell segmentation and tracking in MCF-10A live-cell CDK2 measurements was performed using the H2B-mTurquoise nuclear marker. For segmentation, log-transformed images were convolved with a rotationally symmetric Laplacian of Gaussian and nuclei were defined as contiguous pixels exceeding a threshold score. Tracking of cells from frame to frame was achieved by screening the nearest neighbor for consistency in total H2B-mTurquoise fluorescence. This analysis was performed in Matlab[52,54]. To measure the cytoplasmic:nuclear ratio of DHB-Venus fluorescence, a perinuclear ring with an inner radius 2 μm outside the nuclear mask and an outer radius 10 μm outside the nuclear mask was used for cytoplasmic measurements, excluding regions of the ring within 10 μm of another nucleus. Regions of the ring at or below background fluorescence values were excluded. Global background measurement and subtraction was performed for each channel by dilating the nuclear masks by 50 μm and calculating the median pixel intensity of all non-masked regions. Nuclear fluorescence intensity of DHB-Venus and Geminin-mCherry were calculated as the median nuclear foreground intensity. DHB-Venus cytoplasmic intensity was calculated as the 75th percentile of the foreground of the perinuclear ring. DHB-Venus translocation, our measure of CDK2 activity, was calculated as the ratio of the cytoplasmic signal over the nuclear signal.

**Immunofluorescence**. Cells were fixed in 4% paraformaldehyde then washed three times with PBS. Blocking and permeabilization was then performed using a buffer containing 0.1% triton, 1% BSA, 10% FBS, and 0.01% NaN₃. Primary antibody staining was then performed overnight at 4 °C. Primary antibodies used in this study: phospho-Rb (Ser807/811) (D20B12) Rabbit mAb (Cell signaling technologies #8516, 1:2500); Rb (4H1) Mouse mAb (Cell signaling technologies #9309, 1:1000); p27 Kip1 (D69C12) Rabbit mAb (Cell signaling technologies #3686, 1:1600); p27 Kip1 (SX53G8.5) Mouse mAb (Cell signaling technologies #3698, 1:1000); p21 Waf1/Cip1 (12D1) Rabbit mAb (Cell signaling technologies #2947, 1:2500). Primary antibodies were visualized using a secondary antibody conjugated to AlexaFlour-568 or AlexaFlour-647 (Thermo Fisher, 1:2000) and imaged with a Texas Red or Cy5 filter, respectively.

**Statistics and reproducibility**. An unpaired two-tailed Student's *t*-test was used for analysis in Fig. 4d ($t = 3.53$ df = 4, si-Ctrl vs. si-p21; $t = 4.07$ df = 4, si-p21 vs. si-p27; $t = 0.01$ df = 4, si-Ctrl vs. si-p27) and Supplementary Fig. 4e ($t = 3.58$ df = 4, si-Ctrl vs. si-p21; $t = 4.83$ df = 4, si-p21 vs. si-p27; $t = 0.32$ df = 4, si-Ctrl vs. si-p27). All FRAP curves show mean trace and standard deviation. CDK2 activity traces in Fig. 4c and Supplementary Fig. 4b show median values and 95% confidence intervals. All micrographs are representative images from experiments which were performed on at least two separate days with greater than ten cells imaged in total. All immunoprecipitations and western blots are representative of at least two experiments performed independently on two separate days.

**Reporting summary**. Further information on research design is available in the Nature Research Reporting Summary linked to this article.

## Data availability

The authors declare that the data supporting the findings of this study are available within the paper and its Supplementary information files. Source data are provided with this paper.

## Code availability

Custom cell tracking and analysis code for live-cell experiments is available in a repository on Zenodo https://doi.org/10.5281/zenodo.4701945[54].

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

## Acknowledgements

We thank Charles Sherr for the p21−/− p27−/− MEF cell line and thank Peter Sicinski for the Cyclin D1−/− D2−/− D3−/− MEF cell line; we thank Hee Won Yang for the T-47D cell line; we thank members of the Meyer Lab, Julien Sage, James Ferrell, and Scott Coyle for discussions; we thank Damien Garbett and Anjali Bisaria for assistance with FRAP assays. L.R.P. was supported by National Institute of General Medical Sciences of the National Institutes of Health F32 Ruth L. Kirschstein fellowship F32GM125246. L.H.D. was supported by Stanford Medical Scientist Training Program NIH training grant T32GM007365 and National Institute of Aging of the National Institutes of Health F30 Ruth L. Kirschstein fellowship F30AG060634. T.M. was supported by a National Institute of General Medical Sciences R35 grant (GM127026).

## Author contributions

L.R.P., L.H.D., and T.M. conceived of the project. L.R.P. and L.H.D. designed and executed FRAP and single cell experiments. L.R.P., L.H.D., and M.C. designed and performed the immunoprecipitation assays. L.R.P., L.H.D., and T.M. wrote the manuscript. All authors interpreted data and discussed potential implications throughout the duration of the project.

## Competing interests

The authors declare no competing interests.
