## [Peer Review File · Nature Communications]

Reviewers' Comments:

Reviewer #1:

Remarks to the Author:

The authors have largely addressed the responses to review 1, although importantly the authors have not addressed the point that all work is conducted in a single cell line model. I recommend that the IP of endogenous proteins on p21-CDK4 and p21-CDK2 interaction be conducted in additional cell lines, and ideally should be conducted on estrogen receptor positive cancer cell lines, ie the clinically relevant subtype.

I note the manuscript is elegant biochemistry, and important for this, but is of uncertain clinical translational relevance. No direct evidence is provided to suggest that p21 displacement contributes to palbociclib efficacy in sensitive models. I would advise that this is highlighted in the discussion.

Reviewer #2:

Remarks to the Author:

This is a very important and timely manuscript which addresses a very pressing issue, namely how palbociclib and other CDK4/6 inhibitors inhibit CDK4/6 kinase. This paper refutes the model proposed by Guiley et al in recent study published in Science.

I previously reviewed this manuscript for Nature, and I recommended publication pending clarification of some points that I have raised. The authors have now addressed my concerns. This important study should be published without further delay.

Reviewer #3:

Remarks to the Author:

This remains an important paper with interesting and significant findings on the effects of CDK4/6 inhibitors on the different CyclinD:CDK complexes. The revisions made to the paper have improved the work and I was very pleased to see the inclusion of control experiments for the FRAP studies and IP experiments to confirm their imaging data. The more in-depth examination of the differences between CDK4 and CDK6 was also a nice inclusion and goes some way to explaining the mechanistic differences between p21 dissociation from CDK4 but not CDK6.

I have a few, minor, comments:

- On line 130 the authors state "For all cyclin-CDK complexes, an enrichment could be observed of specific cyclins with their cognate CDKs at the nuclear periphery, indicating that cyclin D can bind CDK4/6 and cyclin E can bind CDK2 and form dimeric complexes in the nuclear environment (Fig. 2a)." However, only images for CyclinD1-CDK4 are shown. The authors should show the images for all of the complexes so that the reader can observe the enrichment.
- In Figure 3H and Supplementary Figure 3H, there are no control experiments showing the dynamics of these proteins in the absence of Palbociclib. These seem like important controls to me as otherwise we don't know if the effects we see in the graphs are specific to Palbociclib treatment.
- While the experiments with Ribociclib in Supplementary Figure 4 b,c show the same trend as for Palbociclib, the difference for Ribociclib is not statistically significant (between si-Ctrl and si-p21). To be absolutely sure this mechanism is broadly applicable to all CDK4/6 inhibitors, perhaps the authors should also try Abemaciclib?
- In the model in Figure 4h, CyclinE/CDK2/p21 complexes also exist pre-Palbociclib addition (as

also seen in Fig 4a) and they will increase on Palbo addition as p21 is relocalised. This should be reflected in the model.

Otherwise, great work!

We thank all of our reviewers for their many positive comments and critiques. The concerns raised by the reviewers were helpful in improving the manuscript. We had added new data and revised the text. Please find our point-by-point responses to the comments below.

Point-by-point reply to the referees' comments:

Specific comments:

Referee #1 (Remarks to the Author):

The authors have largely addressed the responses to review 1, although importantly the authors have not addressed the point that all work is conducted in a single cell line model. I recommend that the IP of endogenous proteins on p21-CDK4 and p21-CDK2 interaction be conducted in additional cell lines, and ideally should be conducted on estrogen receptor positive cancer cell lines, ie the clinically relevant subtype.

We have now performed IPs of endogenous proteins in two estrogen receptor positive cancer cell lines, MCF7 and T-47D. We showed the decrease in CDK4 pull down by p21 in the presence of palbociclib in both cell lines. We showed the increase of CDK2 pull down by p21 in the presence of palbociclib in T-47D, but did not see the same effect in MCF7. As a result we have included additional discussion that the extent of the non-catalytic transfer of p21 to CDK2 will depend on the relative concentrations of CDK4, CDK6, CDK2, and p21.

I note the manuscript is elegant biochemistry, and important for this, but is of uncertain clinical translational relevance. No direct evidence is provided to suggest that p21 displacement contributes to palbociclib efficacy in sensitive models. I would advise that this is highlighted in the discussion.

We agree this is very important to include in the discussion. We have added a sentence to remind readers that we have not tested this in a clinical context and that our results as well as the potential implications as related to clinical data need to be further tested in clinical settings.

Referee #2 (Remarks to the Author):

This is a very important and timely manuscript which addresses a very pressing issue, namely how palbociclib and other CDK4/6 inhibitors inhibit CDK4/6 kinase. This paper refutes the model proposed by Guiley et al in recent study published in Science.

I previously reviewed this manuscript for Nature, and I recommended publication pending clarification of some points that I have raised. The authors have now addressed my concerns. This important study should be published without further delay.

Thank you for your previous comments and support of our manuscript.

Referee #3 (Remarks to the Author):

This remains an important paper with interesting and significant findings on the effects of CDK4/6 inhibitors on the different CyclinD:CDK complexes. The revisions made to the paper have improved the work and I was very pleased to see the inclusion of control experiments for the FRAP studies and IP experiments to confirm their imaging data. The more in-depth examination of the differences between CDK4 and CDK6 was also a nice inclusion and goes some way to explaining the mechanistic differences between p21 dissociation from CDK4 but not CDK6.

I have a few, minor, comments:

- On line 130 the authors state “For all cyclin-CDK complexes, an enrichment could be observed of specific cyclins with their cognate CDKs at the nuclear periphery, indicating that cyclin D can bind CDK4/6 and cyclin E can bind CDK2 and form dimeric complexes in the nuclear environment (Fig. 2a).” However, only images for CyclinD1-CDK4 are shown. The authors should show the images for all of the complexes so that the reader can observe the enrichment.

We have included images for the enrichment of mVenus-Cyclin D1 and mVenus-Cyclin E1 at the periphery in CDK6-mTurq-ΔLamA and CDK2-mTurq-ΔLamA cells, respectively.

- In Figure 3H and Supplementary Figure 3H, there are no control experiments showing the dynamics of these proteins in the absence of Palbociclib. These seem like important controls to me as otherwise we don't know if the effects we see in the graphs are specific to Palbociclib treatment.

Thank you for pointing out that we should include these controls. We should have previously included control experiments for the new section regarding the structural elements that dictate the change in p21 affinity and differences in CDK4 and CDK6. We have now included these in the figures and have incorporated the controls in our descriptions in the text. In adding these controls, we found that the CDK6n-CDK4c does have a lower affinity for p21 compared to wildtype CDK4 and CDK6, and the CDK4n-CDK6c chimera. However, there is still an approximately 10-fold difference in the off-rate comparing the palbociclib treatment and DMSO. We have noted this difference in the text.

- While the experiments with Ribociclib in Supplementary Figure 4 b,c show the same trend as for Palbociclib, the difference for Ribociclib is not statistically significant (between si-Ctrl and si-p21). To be absolutely sure this mechanism is broadly applicable to all CDK4/6 inhibitors, perhaps the authors should also try Abemaciclib?

We have reanalyzed the data to report the change in CDK2 activity rather than percent of cells dropping below an arbitrary threshold. This analysis better accounts for experiment to experiment variability and shows the drop in CDK2 is statistically significant in ribociclib treated cells. We have also included text to suggest further studies comparing these drugs would be warranted.

- In the model in Figure 4h, CyclinE/CDK2/p21 complexes also exist pre-Palbociclib addition (as also seen in Fig 4a) and they will increase on Palbo addition as p21 is relocalised. This should be reflected in the model.

We have updated the model to better reflect the complexes in the cell.

Otherwise, great work!

Thank you for your comments.

Reviewers' Comments:

Reviewer #3:

Remarks to the Author:

The authors have addressed my points and I congratulate them on a really nice piece of research. I look forward to seeing it published.

My only final comment was whether they wanted to broaden the title from just Palbociclib, since they show other clinically-relevant CDK4/6 inhibitors? But this is not essential!